# Roles of DJ41_1407 and DJ41_1408 in *Acinetobacter baumannii* ATCC19606 Virulence and Antibiotic Response

**DOI:** 10.3390/ijms25073862

**Published:** 2024-03-29

**Authors:** Yee-Huan Toh, Guang-Huey Lin

**Affiliations:** 1Master Program in Biomedical Sciences, School of Medicine, Tzu Chi University, Hualien 970374, Taiwan; 110333103@gms.tcu.edu.tw; 2Department of Microbiology and Immunology, School of Medicine, Tzu Chi University, Hualien 970374, Taiwan; 3International College, Tzu Chi University, Hualien 970374, Taiwan

**Keywords:** two-component system, *Acinetobacter baumannii*, transcriptome analysis, virulence, antibiotic response, biofilm, motility

## Abstract

*Acinetobacter baumannii* is a major cause of nosocomial infections, and its highly adaptive nature and broad range of antibiotic resistance enable it to persist in hospital environments. *A. baumannii* often employs two-component systems (TCSs) to regulate adaptive responses and virulence-related traits. This study describes a previously uncharacterized TCS in the *A. baumannii* ATCC19606 strain, consisting of a transcriptional sensor, DJ41_1407, and its regulator, DJ41_1408, located adjacent to GacA of the GacSA TCS. Markerless mutagenesis was performed to construct *DJ41_1407* and *DJ41_1408* single and double mutants. DJ41_1408 was found to upregulate 49 genes and downregulate 43 genes, most of which were associated with carbon metabolism and other metabolic pathways, such as benzoate degradation. MEME analysis revealed a putative binding box for DJ41_1408, 5′TGTAAATRATTAYCAWTWAT3′. Colony size, motility, biofilm-forming ability, virulence, and antibiotic resistance of *DJ41_1407* and *DJ41_1408* single and double mutant strains were assessed against wild type. DJ41_1407 was found to enhance motility, while DJ41_1408 was found to upregulate biofilm-forming ability, and may also modulate antibiotic response. Both DJ41_1407 and DJ41_1408 suppressed virulence, based on results from a *G. mellonella* infection assay. These results showcase a novel *A. baumannii* TCS involved in metabolism, with effects on motility, biofilm-forming ability, virulence, and antibiotic response.

## 1. Introduction

*Acinetobacter baumannii* is a Gram-negative bacterium that can act as an opportunistic pathogen in humans [1], and which is notorious for causing a wide range of severe nosocomial infections [2,3,4]. Among these, ventilator-associated pneumonia and bloodstream infections are regarded as the most significant, with high mortality rates [5]. *A. baumannii* is also categorized as an ESKAPE pathogen, known for high levels of antibiotic resistance [6]. Furthermore, due to its ability to form biofilms, *A. baumannii* can survive on artificial surfaces for prolonged durations, thereby enabling its persistence within hospital settings [7].

*A. baumannii* commonly employs two-component systems (TCSs) to regulate adaptive responses and virulence-related traits. Most TCSs in bacteria can regulate gene expression related to antibiotic resistance, virulence, biofilm formation [8], and motility [9], and 16 sensor kinases and 17 response regulators have been annotated in *A. baumannii* to date, with studies previously conducted on one hybrid sensor kinase, A1S_2811 [10], and nine TCS, AdeSR [11], BaeSR [12], BfmSR [13], CntAB [14], EmaSR [15], GacSA [16], OmpR-EnvZ [17,18], PmrAB [19], and StkSR [20] (AmsSR [21]). Of these TCSs, many were found to regulate gene expression related to antibiotic resistance. For example, AdeSR was found to regulate the expression of the RND-type efflux pump AdeABC in *A. baumannii* BM4454 [22], while BaeSR was shown to regulate the expression of major efflux pumps, including AdeABC, AdeIJK, and MacAB-TolC, to cause decreased tigecycline susceptibility in *A. baumannii* ATCC17978 [23,24]. PmrAB was found to regulate colistin and polymyxin resistance via modification of lipopolysaccharide (LPS) lipid A in both *A. baumannii* ATCC17978 and AB0057 [25,26]. Another function often found to be regulated by TCSs is biofilm-forming ability, and in *A. baumannii* ATCC19606, BfmRS was shown to regulate the *csuA/BABCDE* usher–chaperone assembly system, which produces pili involved in biofilm formation [27], while EmaSR [15] was also found to enhance biofilm-forming ability. Bacterial motility and virulence are often regulated by TCS as well, and BfmRS was shown to regulate expression of the K locus, a critical virulence factor in *A. baumannii* ATCC17978 that includes genes responsible for exopolysaccharide synthesis, including those contributing to capsule production [28]. In addition, EmaSR was found to regulate genes related to motility and virulence in *A. baumannii* ATCC19606 [15], and OmpR-EnvZ (*E. coli* orthologs) was found to be associated with motility and virulence in *A. baumannii* AB5075 [18]. And in *A. baumannii* ATCC17978, A1S_2811 was shown to regulate surface motility through the chaperone/usher pili-associated *csuA/ABCDE* operon [10].

Previous research has revealed that GacSA is a global regulator in *A. baumannii*, controlling the expression of 674 genes associated with virulence, biofilm formation, pili production, resistance against human serum, motility, and the metabolism of aromatic compounds in *A. baumannii* ATCC17978 [29,30]. In *A. baumannii*, GacSA does not exist as a contiguous operon, unlike other well-characterized TCSs such as AdeRS, BaeSR, PmrAB, and BfmRS. Each gene of GacSA is located at different positions on the chromosome; however, the response regulator, GacA, was found to transcribe in the same direction of an adjacent sensor kinase and response regulator of unknown function, termed DJ41_1407 and DJ41_1408, respectively, in *A. baumannii* ATCC19606. In this study, we sought to identify the relationship between DJ41_1407 and DJ41_1408, and understand their functions in *A. baumannii* ATCC19606.

## 2. Results

### 2.1. Exploring the Potential Roles of DJ41_1407, and DJ41_1408 in A. baumannii ATCC19606

In the *A. baumannii* ATCC19606 genome, *gacA*, *DJ41_1407,* and *DJ41_1408* are located adjacently, and have very small intergenic regions, suggesting that they are transcribed as an operon (Appendix A). We were able to amplify the intergenic regions between *gacA* and *DJ41_1407*, and *DJ41_1407* and *DJ41_1408*, using *A. baumannii* ATCC19606 cDNA as a template (Appendix A), and thereby confirming their transcription as a single operon. In addition, in silico analysis predicted a putative histidine kinase domain for self-phosphorylation in DJ41_1407, as well as a putative response regulator domain for phosphorylation in DJ41_1408 (Appendix A), suggesting that they may form a TCS. Notably, in the more commonly studied *A. baumannii* ATCC17978, frameshift mutations in the DJ41_1408 homologue cause its premature termination, and thus the protein is not documented. We therefore elected to investigate DJ41_1407 and DJ41_1408 using the *A. baumannii* ATCC19606 strain.

### 2.2. DJ41_1408 Transcriptome Analysis

Single (*∆DJ41*_*1407*, *∆DJ41*_*1408*) and double (*∆DJ41_1407/08*) mutant strains were developed and confirmed (Appendix A), and comparative analysis of gene expression was conducted between the wild-type and *∆DJ41*_*1408* strains. Transcriptome analysis results were presented in the form of log_2_ expression ratios (wild-type/*∆DJ41*_*1408*), representing the fold-change in expression. A total of 92 genes exhibited expression ratio values greater than 2.0 or less than −2.0. Among these, 49 genes were upregulated, denoted by positive expression ratios (+), while 43 genes were downregulated, indicated by negative expression ratios (−). Gene ontology analysis (Figure 1A) and KEGG classification (Figure 1B) showed that DJ41_1408 predominantly influenced genes related to carbon metabolism and other metabolic pathways, such as benzoate or other aromatic compound degradation (Figure 1C,D). Notably, two gene clusters shown to be upregulated in transcriptome analysis, ranging from 1.4 to 2.8 in terms of log_2_ ratio, were identified as being associated with benzoate degradation (Figure 1C,D). This suggests that DJ41_1408 may be involved in regulating benzoate degradation, but further research will be needed to ascertain this.

### 2.3. Possible Binding Region of DJ41_1408

To identify a possible binding site for DJ41_1408, MEME analysis was conducted to search for regulatory regions within the promoters of differentially expressed genes. A 20-base pair region characterized by the sequence 5′TGTAAATRATTAYCAWTWAT3′ on the positive strand was suggested to be the DJ41_1408 binding site (Figure 2A). This sequence was also found to be consistent with those located in the upstream region of genes with opposite transcriptional orientation. Interestingly, a similar sequence was observed in the upstream region of DJ41_1408 itself, suggesting that it has the capability to regulate its own gene expression, a feature commonly found in most characterized TCSs. The putative DJ41_1408 binding box was identified within the upstream region between positions −130 to −149 of DJ41_1408, displaying a specific sequence of 5′CCTAAACCAGTTCCATTAAT3′ (Figure 2B). However, further research is needed to ascertain the binding characteristics, as well as whether other proteins may be involved in the binding process.

### 2.4. DJ41_1407 and DJ41_1408 Regulate Motility but Not Colony Size

Morphological differences (Figure 3A) between *A. baumannii* ATCC19606 wild-type and mutant strains were analysed with ImageJ 1.53t, revealing similar colony sizes, and indicating that DJ41_1407 and DJ41_1408 do not affect this aspect. A motility assay was also conducted. *A. baumannii* ATCC19606 wild-type and mutant strains were incubated on 0.25% agar plates, and the motility area of each strain was recorded. Upon quantification, the motility areas of *∆DJ41*_*1407* and *∆DJ41*_*1407*/*08* were approximately 1.5-fold smaller than wild-type at both 72 h and 96 h post-incubation, and the results were statistically significant; however, there was no significant difference in the motility area of *∆DJ41*_*1408* compared to wild-type throughout the assay (Figure 3B). The results suggest that DJ41_1407 may be important for the motility of *A. baumannii*.

### 2.5. DJ41_1407 and DJ41_1408 Regulate Biofilm Formation and Virulence

A biofilm formation assay was performed to determine whether DJ41_1407 and DJ41_1408 regulate this capability. *A. baumannii* ATCC19606 wild-type and mutant strains were cultured in 96-well plates for 12 h, and 1% crystal violet was used to stain the biofilm matrix. Results showed that the intensity of crystal violet, which serves as an indicator of biofilm mass, decreased by 1.5-fold in *∆DJ41*_*1408* compared to the wild type, and the results were statistically significant, while *∆DJ41*_*1407* and *∆DJ41*_*1407*/*08* had a similar biofilm mass as the wild type (Figure 4A). This suggests that DJ41_1408 plays a key role in the biofilm-forming ability of *A. baumannii*.

The biofilm-forming ability of bacteria is one of the factors that significantly influences virulence, and to investigate whether DJ41_1407 and DJ41_1408 affect the virulence *of A. baumannii*, an infection assay using *Galleria mellonella* larvae was conducted. *G. mellonella* larvae were infected with 5 × 10^6^ CFU/larva of *A. baumannii* ATCC19606 wild-type and mutant strains, and the survival rate of larvae was recorded every 24 h. At 24 h post-infection, the survival rate of larvae infected with wild-type and *∆DJ41*_*1407* strainsstrain was 50%, while for *∆DJ41*_*1408* and *∆DJ41*_*1407*/*08* strains, the survival rate decreased to 20%. Thereafter, survival rates of larvae infected with wild-type, *∆DJ41*_*1408*, and *∆DJ41*_*1407*/*08* strains remained steady until the end of the 4-day infection period, with no further declines observed. As for larvae infected with *∆DJ41_1407*, the survival rate gradually decreased to 20% upon 3 days post-infection, and then remained steady with no further declines thereafter (Figure 4B). This outcome suggests that DJ41_1407 and DJ41_1408 can play a role in inhibiting the virulence of *A. baumannii*.

### 2.6. DJ41_1408 Modulates Antibiotic Response

To investigate whether DJ41_1407 and DJ41_1408 contribute to *A. baumannii* antibiotic resistance, the minimal inhibitory concentrations (MICs) of different antibiotics to bacterial strains were assessed, using a standardized broth microdilution method, with antibiotic solutions serially diluted 2-fold. Accordingly, wild-type and mutant strains were treated with apramycin, gentamycin, kanamycin, tetracycline, polymyxin B, and colistin. The results of this assay revealed that there was a 2-fold decrease in the MIC for *∆DJ41*_*1408* to gentamicin and kanamycin in comparison to the wild-type strain, while the MIC of all antibiotics for *∆DJ41*_*1407* remained the same as that of the wild-type strain (Table 1). Both kanamycin and gentamicin are aminoglycosides that bind irreversibly to the 30S bacterial ribosome and disrupt protein synthesis, resulting in cell death. These findings suggest that DJ41_1408 is involved in modulating the aminoglycoside response in *A. baumannii* ATCC19606, but further studies will be needed to confirm if DJ41_1407 and DJ41_1408 are involved in antibiotic resistance, and if so, through which underlying mechanisms.

## 3. Discussion

DJ41_1408 was found to modulate the aminoglycoside response in *A. baumannii* ATCC19606, although it is not yet certain whether the effect rises to the level of conferring antibiotic resistance. In *A. baumannii*, several TCSs have already been implicated in conferring resistance to antibiotics. For example, AdeSR enhances resistance to aminoglycosides by regulating the expression of the RND-type efflux pump AdeABC [22]. In other bacteria such as *E. coli*, the CpxAR TCS has been reported to be involved in polymyxin B resistance, even in the absence of the AcrB efflux pump [31]. Further studies are needed to ascertain whether DJ41_1408 can indeed induce aminoglycoside resistance, and if so, to elucidate the underlying mechanisms involved. In *A. baumannii* ATCC19606, it is possible that DJ41_1408 may regulate aminoglycoside resistance through efflux pump-related genes, such as the identified gene cluster DJ41_1868-DJ41_1870, for which the *∆DJ41_1408* strain demonstrated decreased expression levels ranging from 2.4 to 4.2 in terms of log_2_ ratio (Figure 1A). However, further research is needed to understand the detailed mechanisms.

This study showed that mutation of *DJ41_1407* and *DJ41_1408* increases *A. baumannii* ATCC19606 virulence (Figure 4B). In *A. baumannii*, the BfmRS TCS is known to affect virulence via regulation of the *csuA/BABCDE* usher–chaperone assembly system, which produces pili involved in biofilm formation [27], while the GacSA TCS was previously found to regulate the phenylacetic acid (PAA) pathway that is essential for full virulence [30,32]. Conversely, in other bacteria such as *Salmonella enterica*, CpxR/A represses the expression of virulence genes by affecting the stability of the transcriptional regulator HilD [33], while in *Staphylococcus aureus*, the SrrAB TCS may repress virulence under low-oxygen conditions [34]. It is uncertain as to the mechanisms employed by DJ41_1407 and DJ41_1408 to regulate virulence in *A. baumannii* ATCC19606, but from this study, effects on biofilm formation (Figure 4A) were observed, and this may play a role in virulence, for which further investigation is warranted.

Motility assay results showed that DJ41_1407 enhanced the motility of *A. baumannii* ATCC19606 (Figure 3B). In *A. baumannii*, OmpR-EnvZ was found to regulate motility [18], and the hybrid sensor kinase A1S_2811 was reported to regulate surface motility and biofilm formation via the chaperone/usher pili-associated *csuA/ABCDE* operon and the AbaI-dependent quorum-sensing pathway-associated A1S_0112-0119 operon [10]. In other bacteria, such as avian pathogenic *E. coli*, KdpD/KdpE contributed to the expression of some flagella-related genes involved with flagella formation and motility [35], while in *Dickeya oryzae*, ArcBA was found to modulate cell motility and biofilm formation [36]. Interestingly, the recent research in *A. baumannii* strain MAR002 has shown that virulence is decreased when impairments in surface-associated motility occur, but not when twitching motility is affected [37]. It is uncertain as to the type(s) of motility affected by DJ41_1407 in *A. baumannii* ATCC19606, and the exact mechanisms involved in motility increase and the implications for virulence and persistence require further investigation.

In conclusion, this study identified a novel sensor (DJ41_1407) and regulator (DJ41_1408) located in the same transcript as GacA in *A. baumannii* ATCC19606, and demonstrated that this TCS can influence motility, biofilm formation, virulence, and antibiotic resistance.

## 4. Materials and Methods

### 4.1. Bacteria Strains, Plasmids, and Media

The bacterial strains used in this study are listed in Table 2, while the plasmids and primers used in mutant construction are listed in Table 3 and Table 4. LB (lysogeny broth) medium and agar were used to grow bacterial strains either statically or with shaking at 200 rpm at 37 °C.

### 4.2. Markerless Mutation

PCR was performed using the corresponding primers to amplify the upstream 1 kb and downstream 1 kb of DJ41_1407 or DJ41_1408, respectively (Appendix A). The upstream and downstream 1 kb fragment were then each inserted into the suicide plasmid pK18*mobsacB*, which was then transformed into *E. coli* S17-1λπ. The derived recombinant plasmid was subsequently transformed into *A. baumannii* through conjugation. Culturing on LB plates containing 50 μg/mL ampicillin and 50 μg/mL kanamycin was performed to select *A. baumannii* with the pK18*mobsacB* plasmid integrated into chromosomes, indicative of first homologous recombination. Colony PCR was then performed to amplify the upstream to downstream fragment for selection of colonies that successfully underwent 1st homologous recombination. Selected colonies were then cultured in LB medium containing 20% sucrose. SacB protein translated from the pK18*mobsacB* plasmid digests sucrose into leaven, which is lethal for bacterial survival, and therefore 2nd homologous recombination will occur, with 50% possibility for the deletion of gene fragments to be mutated [41]. Mutation confirmation results are shown in Appendix A.

### 4.3. Transcriptome Analysis Preparation

Bacterial strains were cultured in LB medium containing 50 μg/mL ampicillin at 37 ℃ for 12–14 h, and then subcultured with initial OD_600_ of 0.1 for 3 h. After 3 h, the bacterial solution was aliquoted to OD_600_ of 0.6 for every microcentrifuge tube, and centrifuged at 13,000× *g* for 10 min. After centrifugation, the supernatant was removed, and the pellet was resuspended with 1 mL Thermo Fisher TRIzol™ Reagent (Invitrogen, CA USA). After resuspension, the sample was stored at −80 °C and then sent to Welgene Biotech Co., Ltd., (Taipei, Taiwan) for transcriptome analysis.

### 4.4. Galleria Mellonella Larvae Infection Assay

Bacterial strains were cultured in LB medium at 37 °C for 12–14 h. The bacterial solution was then washed twice with phosphate-buffered saline buffer (PBS, 0.14 M NaCl, 2.7 mM KCl, 8.1 mM Na_2_HPO_4_, 1.5 mM KH_2_PO_4_) to remove LB medium, and then diluted to 5 × 10^8^ CFU/mL. Larvae that were infected only with PBS and heat-killed wild-type served as the control group (n = 10). Heat-killed wild-type was prepared by heating the bacterial solution at 100 °C for 5 min. Using Hamilton syringes [Hamilton^®^ syringe, 700 series, 701N], 10 mL of diluted bacterial solutions was injected into *G. mellonella* larvae through the last left pro-leg of the abdominal region. The final bacterial concentration was 5 × 10^6^ CFU/larva. Infected larvae were incubated at 37 °C. Scores for survival (alive/dead) and melanisation were measured every 24 h for 96 h [42].

### 4.5. Biofilm Formation Assay

Bacteria were cultured at 37 °C for 12–14 h. The optical density of overnight cultures was measured and adjusted to OD_600_ = 1. Bacteria were then cultured in 96-well plates from OD_600_ = 0.1 for 12–14 h, after which 1% crystal violet was added into the bacterial solution and stained for 30 min. All bacterial solutions were removed from each well and rinsed with ddH_2_O twice. The 96-well plate was allowed to dry for 15 min. The stained biofilm in each well was dissolved with 300 mL 95% ethanol for 30 min and measured with OD_595_ (Thermo Fisher Scientific, Waltham, MA USA; Multiskan^TM^ SkyHigh) [43].

### 4.6. Motility Assay

Overnight bacterial culture was adjusted to OD_600_ = 1.0, and 2 mL of bacterial solution was dropped at the centre of a 0.25% LB agar plate. The LB plate was incubated at 37 °C in the dark, and images were taken at 24 h-intervals for at least 96 h. Motility areas were analysed with ImageJ 1.53t [10].

### 4.7. MIC Test

The antibiotic resistance of each strain was assessed by liquid minimum inhibition assay [39,44]. Antibiotics were inoculated into 96-well microtiter plates with 2-fold serial dilution. The strains were cultured in 3 mL of LB medium at 37 °C overnight. Diluted overnight cultures with an OD_600_ of 0.1 were then inoculated into 96-well microtiter plates. Optical density was measured after overnight culture, and the MIC was defined as the lowest antibiotic concentration inhibiting bacterial proliferation. Bacterial strains were treated with apramycin, gentamycin, kanamycin, tetracycline, polymyxin B, and colistin. Each desired antibiotic experimental group was serially performed with six replicates [39].

## Figures and Tables

**Figure 1 ijms-25-03862-f001:**
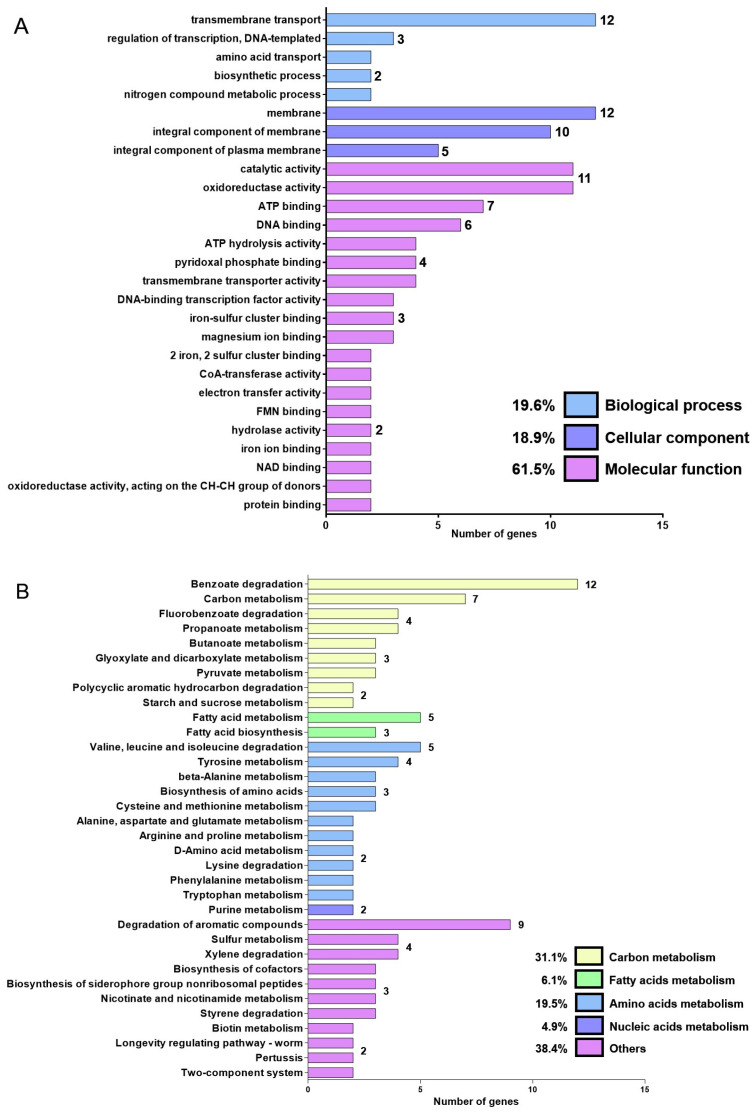
Transcriptome analysis of differential gene expression between the *A. baumannii* ATCC19606 wild-type and DJ41_1408 mutant strains. (**A**) Differential gene expression by gene ontology classification, in which only clusters with more than 2 genes are displayed. (**B**) Differential gene expression by KEGG classification, in which only clusters with more than 2 genes are displayed. (**C**) Fold-change in gene expression for DJ41_332-DJ41_340 and DJ41_2252-DJ41_2255, gene clusters involved in benzoate degradation. Black lines indicate individual transcriptional units, and white or grey arrows represent transcription direction. (**D**) DJ41_332-DJ41_340 and DJ41_2252-DJ41_2255 annotation in the *A. baumannii* ATCC19606 benzoate degradation pathway. Black text: gene number; red text: gene name; blue text: log_2_ expression ratio (wild-type/*∆DJ41_1408*).

**Figure 2 ijms-25-03862-f002:**
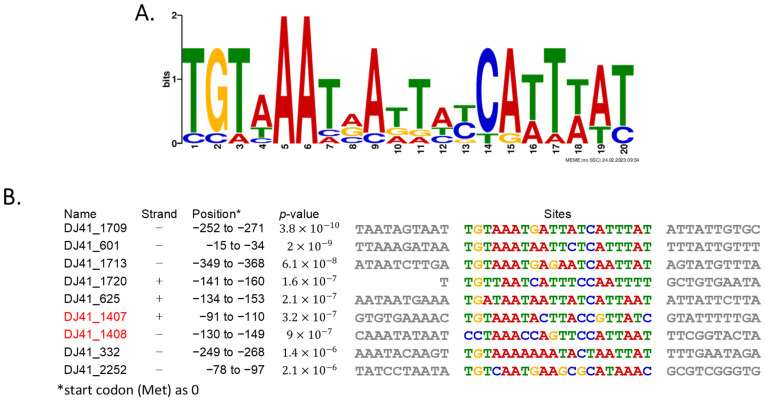
Putative DJ41_1408 binding box analysis by MEME. (**A**) Conservation of the DJ41_1408 binding box. (**B**) The exact sequence of the DJ41_1408 binding box in promoter regions of different genes, which were analysed using MEME to identify highly conserved sequences. Positions represent the location of the conserved sequence at the upstream of analysed genes, and the first base pair of the start codon was considered as +1. The highly conserved sequence may be the binding site of DJ41_1408. MEME link: https://meme-suite.org/meme/tools/meme (accessed on 13 April 2023).

**Figure 3 ijms-25-03862-f003:**
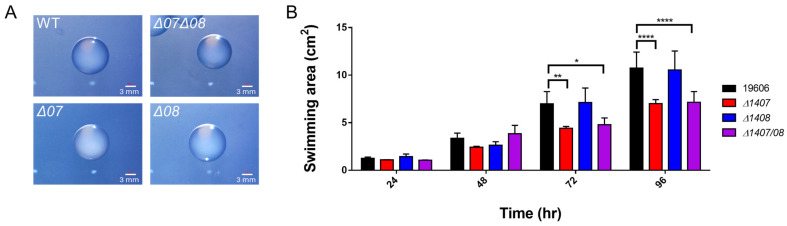
(**A**) Photographs of *A. baumannii* ATCC19606 wild-type and mutant strain colonies were taken at 12 h post-cultivation. The white scale bar represents 3 mm. ImageJ 1.53t was used to analyse the diameter of colonies, using the scale bar as a standard, and with 10 colony replicates measured in each strain. (**B**) Measured motility area of *A. baumannii* ATCC19606 and mutant strains. Photographs were taken at 72 h post-cultivation, and ImageJ 1.53t was used to analyse the motility area of all strains. * *p* < 0.05; ** *p* < 0.01; **** *p* < 0.001.

**Figure 4 ijms-25-03862-f004:**
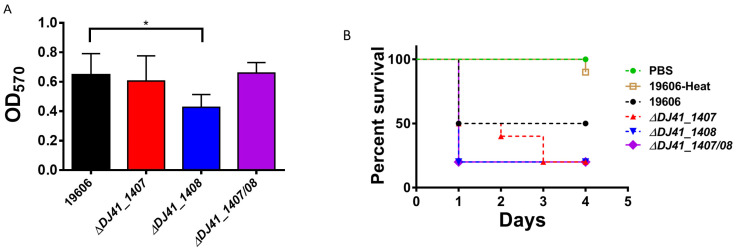
(**A**) Biofilm-forming ability of *A. baumannii* ATCC19606 wild-type and mutant strains. OD_570_ represents biofilm formation ability. * *p* < 0.05. (**B**) *G. mellonella* survival rates after infection with *A. baumannii* ATCC19606 wild-type and mutant strains. Kaplan–Meier survival curves are shown, with each curve representing a single representative experiment performed with 10 larvae.

**Table 1 ijms-25-03862-t001:** Minimal inhibitory concentrations (MICs) of different antibiotics against bacterial strains.

STRAIN	APR	GEN	KAN	TET	PMB	CST
Wild-type	50	100	25	0.78	6.25	5
*∆DJ41_1407*	50	100	25	0.78	6.25	5
*∆DJ41_1408*	50	** 50 **	** 12.5 **	0.78	6.25	5
*∆DJ41_1407∆DJ41_1408*	50	** 50 **	** 12.5 **	0.78	6.25	5

APR: apramycin, GEN: gentamicin, KAN: kanamycin, TET: tetracycline, PMB: polymyxin B, CST: colistin. **Bold underline**: A 2-fold lower MIC in comparison to wild type. The unit of antibiotics used was mg/mL.

**Table 2 ijms-25-03862-t002:** Bacterial strains used in this study.

Bacteria	Description	References or Source
*E. coli*		
DH5α	F^−^, *supE44*, *hsdR17*, *recA1*, *gyrA96*, *endA1*, *thi-1*, *relA1*, *deoR*, λ	ATCC53868
DH5α/pK18-*∆DJ41_1408*	Kan^r^, DH5α containing pK18_*∆DJ41_1408*	This study
DH5α/pK18-*∆DJ41_1407-08*	Kan^r^, DH5α containing pK18-*∆DJ41_1407-08*	This study
S17-1λπ	*thi-1*, *thr*, *leu*, *tonA*, *lacY*, *supE*, *recA*, *RP4-2* (*Km::Tn7,Tc::Mu-1*), *Smr*, *lpir*	[38]
S17-1λπ/pK18-*∆DJ41_1407*	Kan^r^, S17-1λπ containing pK18_*∆DJ41-1407*	[39]
S17-1λπ/pK18-*∆DJ41_1408*	Kan^r^, S17-1λπ containing pK18_*∆DJ41_1408*	This study
S17-1λπ/pK18-*∆DJ41_1407-08*	Kan^r^, S17-1λπ containing pK18-*∆DJ41_1407-08*	This study
*A. baumannii*		
ATCC19606	Amp^r^, clinical isolate, wild type	[40]
*∆DJ41_1407*	Amp^r^, deletion of *DJ41_1407*	[39]
*∆DJ41_1408*	Amp^r^, deletion of *DJ41_1408*	This study
*∆DJ41_1407∆DJ41_1408*	Amp^r^, deletion of *DJ41_1407* and *DJ41_1408*	This study

**Table 3 ijms-25-03862-t003:** Primers used in this study.

Primer	Sequence (5′-3′)	Application	Reference or Source
pK18-DJ41_1408UP_F	AATTCGAGCTCGGTACCCGGGATATTAACCGTTATATCTTA	construct and confirm mutant	This study
pK18-DJ41_1408DO_R	GTAAAACGACGGCCAGTGCCACGTCCGTGATAACTATGTCG	construct and confirm mutant	This study
DJ41_1408UP-DJ41_1408DO_F	AAATGGCAGAACAGCAACCGGAGACAGAACAGGAAGTA	construct mutant	This study
DJ41_1408DO-DJ41_1408UP_R	TACTTCCTGTTCTGTCTCCGGTTGCTGTTCTGCCATTT	construct mutant	This study
pK18-DJ41_1407UP_F	GAGCTCGGTACCCGGGGTGGTTGAGAACTGACGAAT	construct and confirm mutant	This study
DJ41_1408DO-DJ41_1407UP_R	TACTTCCTGTTCTGTCTCCGACGAAAAATCGTATGGGACA	construct mutant	This study
DJ41_1407UP-DJ41_1408DO_F	TGTCCCATACGATTTTTCGTCGGAGACAGAACAGGAAGTA	construct mutant	This study
1406_up380-F	GAGCTCGGTACCCGGGACGGGTTATTGACGAGTTCT	confirm mutant	This study
1408_do250-R	GTGCTTGGGTTATGGGTGAA	confirm mutant	This study
AbEraS+R_intergenic region-qF	GCGATTCGTTAC GGTTTGAT	amplify intergenic region	[39]
AbEraS+R_intergenic region-qR	AGGAATATAAGGCAGGTTGCTG	amplify intergenic region	[39]
1407/08-IGR-qF	GGGGCATGCTTCAGAATAGA	amplify intergenic region	This study
1407/08-IGR-qR	TAATCCCCATTCGTGCAAGT	amplify intergenic region	This study

**Table 4 ijms-25-03862-t004:** Plasmids used in this study.

Plasmids	Description	References or Source
pK18mobsacB	Kan^r^, mobilizable suicide vector, *sac*B, *ori*T	[41]
pK18-*∆DJ41_1407*	Kan^r^, pK18*mobsacB* containing *DJ41_1407* upstream and downstream 1 kb fragments	[39]
pK18-*∆DJ41_1408*	Kan^r^, pK18*mobsacB* containing *DJ41_1408* upstream and downstream 1 kb fragments	This study
pK18-*∆DJ41_1407-08*	Kan^r^, pK18*mobsacB* containing *DJ41_1407* and *DJ41_1408* upstream and downstream 1 kb fragments	This study

## Data Availability

The data for this study are available upon reasonable request to the corresponding author.

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
