# Peer review of "Roles of DJ41_1407 and DJ41_1408 in Acinetobacter baumannii ATCC19606 Virulence and Antibiotic Response"

_ijms, 2024, doi:10.3390/ijms25073862_

Round 1

Reviewer 1 Report

Comments and Suggestions for Authors

Although the research described in this manuscript appears to be well done. It is difficult to make a through assessment because of the quality of the English. 

Comments on the Quality of English Language

Examples of the kinds of changes that need to be made are shown below:

Line 84 -85 rewrite “..DJ41_1408 are three adjacent genes that are transcribed in the same direction …”

Line 86-88 Change to “Therefore, we hypothesized that the three genes were transcribed as an operon with a single mRNA that coded for all three proteins. To test this hypothesis, RNA was extracted …””

Author Response

Thank you very much for taking the time to review this manuscript. Please find the detailed responses in the attachment below.

Reviewer 2 Report

Comments and Suggestions for Authors

In this study, the authors characterize a novel TCS in A. baumannii. Deleting the gene led to mis-regulation of gene transcription, specifically in genes involved in benzoate metabolism. The authors predict the binding site for the 1408 regulator. They carry out studies to characterize the mutant strains, and report increase in virulence, and a small increase in biofilm formation, motility, and antibiotic resistance.  

The results are of potential interest, and I appreciate the large amount of work performed by the authors. However, the study suffers from several crucial flaws in the experimental design and uses inappropriate methodology. Moreover, the results are presented in such an unclear way that it is almost impossible to interpret their meaning. The transcriptome analysis probably yielded interesting results – but those are either not presented or not tested – and definitely not discussed…

I would suggest performing the crucial controls, amend the methodology as suggested below, but mostly – streamline the manuscript and present only meaningful results, with an appropriate discussion of the main findings.

Major:

1.       Revise English in the results section.

2.       The detailed descriptions of the figures belong in figure legend – not in the results section. For example, L91-94:

"In Figure 1B, lane 1 is the DNA marker, lane 2 is a negative control without any template, lane 3 is the A. baumannii chromosome as template (Positive control), and lane 4 is the A. baumannii cDNA as template."

3.       It is clear from the DNA architecture that the 3 genes constitute a single operon, and I don’t think showing this experimentally adds a lot. I would remove this result (2.1) altogether, or maybe mention it as one sentence and move the figure to supplementary material. However – if the result remains in the article, the experiment is missing a critical control. A negative control (cDNA reaction without reverse transcriptase added) needs to be performed, to demonstrate that the amplified region is not due to residual DNA left in the RNA prep.

4.       The description of the predicted domain analysis is very long, and unless the authors test these predictions experimentally, adds little. The two genes were previously annotated as sensor kinase and response regulator, most likely based on the predicted domains. I see no evidence that the two proteins act together (the main claim of this section " DJ41_1407 and DJ41_1408 form a TCS in A. baumannii ATCC19606") – this needs to be established experimentally. I would significantly shorten this section (2.2)

5.       It is interesting that the two genes are mutated and missing in ATCC 17978 strain. However, again – this observation does not warrant 15 lines (section 2.3). It is enough to mention that there is a frameshift eliminating the protein. The detailed mention of all the different mutations adds nothing. Moreover – as the authors had a natural "null" mutant, why didn't they use it as a control in the following experiments?

6.       The transcriptome results (2.4) may be very interesting – but they are presented in a very confusing way. The fact that most mis-regulated genes are "molecular processes" is not meaningful by itself. L155-165 should be removed – they are just describing the very clear figure and add nothing. This whole section is a long list of results, and their meaning is not clear. Why is it important to detail the processes represented by a single gene (Fig. 2A/B). Those results should be presented as a supplementary table. The only meaningful finding of this section (2.5) I could find is regulation of benzoate metabolism (and maybe membrane transport – but this one is completely lost in the unnecessary details and is not discussed further). There is absolutely no reason to describe the benzoate degradation pathway in 20 lines (L177-L197) – those are not the findings of this paper. Instead, to make this claim, experiments directly testing benzoate metabolism in the wt and the mutants have to be conducted.

7.       The same goes for the potential binding site. It is an interesting and novel result – but should be tested experimentally, by site directed mutagenesis and a reporter gene – or but a direct protein binding assay.

8.       The colony size did not differ between the mutant and the wt. Why is it described in 6 lines and two figure panels?

9.       The motility and the biofilm assays yielded a small, but significant difference. The authors should discuss the possible genes mis-regulated in the mutant that might be involved in those processes in the Discussion section. The same goes for the virulence assay, where the results are even more dramatic

10.   MICs were determined by a completely unacceptable method. LB medium should not be used, and the concentrations can not be randomly chosen! The results achieved in this way are impossible to interpret. MIC should be determined using standard and validated clinical methods, following EUCAST or CLSI guidelines.

11.   Discussion is not clear, hard to follow, and is not really related to the actual findings of the paper. It mainly lists other TCS which were found in other works to regulate the processes tested in this paper. For example: Did any of the efflux pumps involved in aminoglycoside were affected in the transcriptome analysis conducted in this work? If so – which? This is not clear from the transcriptome results, and instead of discussing their own findings on the matter – the authors spend 35(!) lines on a lengthy description of other TCS in other bacteria. Why resistance to beta-lactams is discussed at such length?

Comments on the Quality of English Language

A minor revision of English (especially in the Results section) is suggested.

Author Response

(The authors gave the same response as above.)

Round 2

Reviewer 1 Report

Comments and Suggestions for Authors

This manuscript can now be read and evaluated. The analysis seems straightforward and the conclusions appear to be justified.

Comments on the Quality of English Language

This manuscript is much better than the original submission, but their are still many places where the English can be improved.

Author Response

Dear Editor,

Thank you very much for facilitating the review of this manuscript, and we thank the reviewers for their insights and suggestions for revision, which have greatly enhanced our manuscript. In response to the reviewer comment regarding minor modification of English, we have edited our manuscript again, and have listed the main changes made in the Response to Reviewers below.

Hopefully, these changes will further enhance the readability of this manuscript, and facilitate your editorial decision. Thank you very much for taking the time to review these materials, and we look forward to your feedback.

Sincerely,

Dr. Guang-Huey Lin

Department of Microbiology and Immunology, School of Medicine; and International College,
Tzu Chi University, Hualien 970374, Taiwan

Reviewer 2 Report

Comments and Suggestions for Authors

The authors made a sincere and mostly successful effort to address my previous concerns, and the manuscript is now significantly improved, much clearer and easier to follow. The finding of regulation of carbon metabolism, motility, and virulence by DJ41_1407/8 TCS are of interest. 

I have some suggestions, and one remaining major concern, which (in my view) absolutely must be addressed before publication. 

Major:  

The role of DJ41 in antibiotic resistance 

I understand that the authors did not "aim to obtain clinically relevant MIC". However, to claim that any of the results are relevant to the organism's resistance, they must be presented in the correct context. The authors make a significant claim, that DJ41_1408 regulates "antibiotic, especially aminoglycoside" resistance. I remain unconvinced this is true, and I do not think their results show this. 

How are they deciding that the change in MIC is significant? The change in MIC must be compared to something…  

I could find no clinical breakpoints for Apramycin or Kanamycin for A. baumannii. Therefore, I do not know whether, for Kanamycin, the resistant WT (MIC 25) now became sensitive mutant (MIC 12.5), remained resistant mutant, or the WT was sensitive to begin with… For other Enterobacterales, an organism is considered resistant with MIC >64. Therefore, if we use this as a reference, a sensitive bacterium remained sensitive. As for Gentamycin, A. baumannii is considered resistant when MIC is above 16 mg/ml. So, in this case, the extremely resistant A. baumannii WT remained extremely resistant mutant. This is a minor change in sensitivity to antibiotics, and its significance is limited.  

Even if not compared to clinical MIC breakpoints, to make any resistance claims, the authors must find a reference point (even if it is a research article) – otherwise the whole claim is not convincing. 

Crucially, in my experience, a two-fold difference in MIC can be due to natural variation of the bacterial growth, inoculum effect, the medium used, the batch and age of antibiotic used, and much more. Deletion of a gene significantly involved in antibiotic resistance would change the MIC much more than by a single dilution. From the Materials and Methods section, it seems that the experiment was conducted only once, with 6 technical replicates, and without any controls. No information is given about the variation between the repeats. No biological repeats were conducted. This is simply not enough to substantiate the authors claim. 

Even more frustrating is the part of the Discussion referring to the antibiotic resistance (L205-218): 

" In other bacteria such as E. coli, CpxAR TCS is involved in polymyxin B resistance, as overexpression of CpxR can result in resistance to beta-lactam antibiotics, even in the absence of the AcrB efflux pump [31]. 

  • Polymyxin is not a beta-lactam, and as far as I can tell, reference 31 is not related to either lipopeptides or beta-lactams. 

"This finding is significant because bacteria can initiate a response to β-lactam antibiotics even before the bacterial cell wall is damaged by these compounds." 

  • There is no finding, the authors did not test any beta-lactams, and the reference does not mention beta-lactams.  

The claim about antibiotic resistance should be very carefully reconsidered and revised, so as not to overstate the significance of the finding. And it must be put in context, with a meaningful discussion.  

Minor: 

  1. The significance is presented in a very unusual and confusing way, in Figure 3B and Figure 4A. The authors use letters to state which results are NOT significantly different from each other. Instead, they should use the conventional asterisk, choose one point of reference, and only mark the results for which the difference IS statistically significant from the reference. P values must be stated. 

  1. L22 – "Double" is missing? 

  1. Figure 1 – I again suggest removing the categories represented by a single gene from Figure 1. This would make it much clearer and more focused. 

  1. L103 – remove "highly" – 2.8-fold is not high. 

  1. Lines 173 – a typo in the name of the mutant. Also, the text describing this figure is extremely confusing, as when you state "A, B and C remained the same", I understand that there was no difference between them… 

  1. Remove L189-190. This is clear, and stating this does not resolve the problem with this section. 

  1. L198 – remove "especially regarding resistance to aminoglycosides". Instead, the first part should read "modulating aminoglycoside resistance". 

  1. L 220-222 – seem not related to the rest of the paragraph. 

  1. In general, in the Discussion, I would place more emphasis on how the genes diss-regulated in this study could regulate/affect the phenotypes of the mutant – and less on how completely unrelated TCSs in other bacteria regulate similar processes. 

Author Response

Dear Editor,

Thank you very much for facilitating the review of this manuscript, and we thank the reviewer for offering detailed and specific suggestions regarding the improvement of our manuscript. We provide our response below and have implemented the required changes into our revised manuscript accordingly. Hopefully, these changes will be able to assist your editorial decision, and we offer our sincere thanks for the time and effort spent in the review of our manuscript.

Sincerely,

Dr. Guang-Huey Lin

Department of Microbiology and Immunology, School of Medicine; and International College,
Tzu Chi University, Hualien 970374, Taiwan
